# Dietary Glycemic Index and Glycemic Load Are Not Associated with the Metabolic Syndrome in Lebanese Healthy Adults: A Cross-Sectional Study

**DOI:** 10.3390/nu12051394

**Published:** 2020-05-13

**Authors:** Cecile Borgi, Mandy Taktouk, Mona Nasrallah, Hussain Isma’eel, Hani Tamim, Lara Nasreddine

**Affiliations:** 1Department of Nutrition and Food Sciences, Faculty of Agricultural and Food Sciences, American University of Beirut, Beirut P.O.BOX: 11-0236, Riad El Solh 1107-2020, Lebanon; ce37@aub.edu.lb (C.B.); mt86@aub.edu.lb (M.T.); 2Department of Internal Medicine, Division of Endocrinology, Faculty of Medicine, American University of Beirut Medical Center, Beirut P.O.BOX: 11-0236, Riad El Solh 1107-2020, Lebanon; mn36@aub.edu.lb; 3Vascular Medicine Program, American University of Beirut Medical Center, Beirut P.O.BOX: 11-0236, Riad El Solh 1107-2020, Lebanon; hi09@aub.edu.lb; 4Division of Cardiology, Department of Internal Medicine, Faculty of Medicine, American University of Beirut Medical Center, Beirut P.O.BOX: 11-0236, Riad El Solh 1107-2020, Lebanon; 5Department of Internal Medicine, Faculty of Medicine, American University of Beirut Medical Center, Beirut P.O.BOX: 11-0236, Riad El Solh 1107-2020, Lebanon; 6Clinical Research Institute, Faculty of Medicine, American University of Beirut Medical Center, Beirut P.O.BOX: 11-0236, Riad El Solh 1107-2020, Lebanon

**Keywords:** glycemic index, glycemic load, metabolic syndrome, urban adults, Lebanon

## Abstract

High dietary glycemic index (GI) and glycemic load (GL) were suggested to increase the risk of metabolic syndrome (MetS). This study aims to estimate dietary GI and GL in a sample of healthy Lebanese adults and examine their association with MetS and its individual abnormalities. The study uses data from a community-based survey of 501 Lebanese urban adults. Dietary intake was assessed using a food frequency questionnaire. Biochemical, anthropometric, and blood pressure measurements were obtained. Subjects with previous diagnosis of chronic disease, metabolic abnormalities, or with incomplete data or implausible energy intakes were excluded, yielding a sample of 283. Participants were grouped into quartiles of GI and GL. Multivariate logistic regression analyses were performed. Average dietary GI and GL were estimated at 59.9 ± 8 and 209.7 ± 100.3. Participants belonging to the highest GI quartile were at increased risk of having MetS (odds ratio (OR) = 2.251, 95% CI:1.120–4.525) but this association lost significance with further adjustments. Those belonging to the second quartile of GI had significantly lower odds of having hyperglycemia (OR: 0.380, 95% CI:0.174–0.833). No associations were detected between GL and MetS. The study contributes to the body of evidence discussing the relationship between GI, GL, and MetS, in a nutrition transition context.

## 1. Introduction

The metabolic syndrome (MetS) refers to a constellation of cardiometabolic risk factors that identify individuals at particularly high risk for cardiovascular disease and diabetes mellitus [1]. These risk factors include elevated fasting blood glucose (FBG), raised blood pressure, elevated serum triacylglycerols (TAG) levels, low high-density lipoprotein (HDL) cholesterol levels, and obesity, particularly central adiposity [2]. The MetS has been recognized as a major public health concern putting more than a billion people at an increased risk of morbidity and mortality worldwide [3,4]. Several genetic and environmental factors have been proposed as potential drivers for MetS development [5,6]. Among those, diet is suggested as a potential modifiable risk factor in MetS etiology, with a plethora of studies having investigated the effect of single food times, nutrients, or dietary patterns on MetS risk [3,7,8,9,10]. In this context, increasing attention has been accorded to carbohydrate intake, especially from refined grains and sugars [11]. The resulting glycemic response, expressed as the postprandial change in blood glucose level, was suggested as a crucial determinant of cardiometabolic risk in a recent scientific consensus held by the International Carbohydrate Quality Consortium [12]. This postprandial glycemic response is dictated by both the quantity of carbohydrates ingested and the rate of its gastrointestinal absorption, which may vary between different types of carbohydrates. In this context, the glycemic index (GI) and the glycemic load (GL) were proposed as measures of carbohydrates’ quality and quantity, respectively. The GI is based on the incremental blood glucose response and insulin demand for a given amount of carbohydrate [13]. It ranks carbohydrate-rich foods based on how much they raise blood glucose concentrations compared with a reference food [14]. However, the GI does not take into consideration the amount of carbohydrates in the diet, it only addresses the quality of the carbohydrate. Yet, the quantity of dietary carbohydrate is also an important factor in determining the postprandial glucose response [14]. Hence, the concept of dietary GL was developed [15]. The GL is determined as the product of a food’s GI and its total available carbohydrate content [14].

Few studies have examined the association between dietary GI, GL, and MetS, yielding conflicting results in various populations [16]. The Framingham Offspring Cohort detected a positive association between GI and MetS in the population as a whole [17], while the Prevención con Dieta Mediterránea (PREDIMED) study suggested that this association may be age-dependent with an increased MetS risk observed in younger age groups but not in elderly subjects [18]. Data from the National Health and Nutrition Examination Survey (NHANES) III did not detect any association between GL and MetS [19], while the Cooper Center longitudinal study showed that, contrary to expectations, men belonging to the highest quintile of GL were at a decreased risk of developing MetS [20].

In the Arab Eastern Mediterranean Region (EMR), studies investigating the association between GI, GL, and MetS are lacking, although the region has witnessed dramatic increases in the prevalence of MetS during the past few years [21,22]. Within the region, Lebanon has been on the lead [21,22], with a reported MetS prevalence of 34.7% among adults [23]. This high prevalence of MetS may be a reflection of the alarming escalation of obesity rates in Lebanon [24] while also being a reflection of the significant shifts in food consumption patterns that the country has witnessed during the past few decades [25]. Like other countries of the EMR, Lebanon is undergoing the nutrition transition with its characteristic shifts in food supply and diet, away from the traditional pattern that was rich in whole grains, legumes, fruits, and vegetables, toward a “westernized” pattern that is rich in refined grains, added sugar, and animal-based products [25]. It is in this context that the present study was undertaken, with the aims of evaluating the association between dietary GI, GL, and cardiometabolic abnormalities in healthy Lebanese adults. Based on a cross-sectional survey conducted in 2014, the specific objectives of the study are to (1) determine the GI and GL of Lebanese food items based on pertinent literature; (2) estimate daily dietary GI and GL; (3) examine the association of dietary GI and GL with fasting blood lipid levels, fasting glycemia, and blood pressure; and (4) investigate the association between dietary GI and GL and the MetS in a sample of healthy Lebanese adults. Findings of this study can be used to develop culture-specific, evidence-based intervention strategies that contribute to better cardiometabolic health among Lebanese adults.

## 2. Materials and Methods

This is a cross-sectional study based on a community-based survey conducted on a representative sample of Lebanese adults residing in the Greater Beirut area. The survey was conducted between March and May 2014. All subjects gave their informed consent for inclusion before they participated in the study. The study was conducted in accordance with the Declaration of Helsinki, and the protocol was approved by the Institutional Review Board (IRB) of the American University of Beirut (Project identification code: IM.HT.03).

### 2.1. Study Participants

The study design and sampling strategy are published elsewhere [26,27,28]. In brief, a representative sample of Lebanese adults was recruited based on a multistage stratified probability sampling frame [29]. Within this frame, the strata were the districts of the Greater Beirut area. Within each district, neighborhoods and households were selected based on systematic random sampling. The numbers of neighborhoods and households selected within each district were proportional to the total numbers of neighborhoods and households within the district, respectively. At the household level, when more than one adult respondent was eligible to participate, information on the month of birth was obtained and the adult respondent with the most recent month of birth was recruited [26,27,28].

The inclusion criteria were age above 18 years, of Lebanese nationality, and residing in Greater Beirut. Participants were excluded if they were on dialysis, mentally disabled, or pregnant. In addition, given that the parent study investigated occupational exposure to Bisphenol A (BPA) [26], subjects who worked in plastic or chemical factories were excluded. Eligible participants who agreed to participate in the survey were invited to the Department of Nutrition and Food Sciences at the American University of Beirut for data collection. The study population included 501 participants who participated in the original survey. For the present study, the selection of participants from the original survey population (*n* = 501) was performed according to the following criteria: (i) having complete dietary, anthropometric, blood pressure, and biochemical data and (ii) with no previously known diagnosis of chronic disease or any metabolic abnormality, yielding 314 participants. In addition, subjects who reported implausible energy intakes (<500 or > 6000 kcal/day) were excluded [11,16]. Accordingly, 283 participants were included in the current analysis.

### 2.2. Data Collection

Subjects were asked to visit the Department of Nutrition and Food Sciences after an overnight fast (10 h). The study participants completed a multicomponent questionnaire and a food frequency questionnaire (FFQ) in a one-on-one interview setting. Anthropometric measurements (weight, height, and waist circumference) were obtained and blood samples were drawn by a licensed phlebotomist. The duration of the interview and data collection for each participant was approximately one hour.

#### 2.2.1. Sociodemographic, Anthropometric, and Biochemical Assessment

Using a sociodemographic and lifestyle questionnaire, information about the following criteria were collected: age (in years), gender, monthly income (expressed in U.S dollars), marital status, education, smoking status, sleeping difficulties, alcohol intake, and personal and family medical history (coronary artery disease, hypertension, diabetes mellitus, dyslipidemia, thyroid disease, cancer). The short version of the International Physical Activity Questionnaire (IPAQ) was adopted as an interviewer-administered tool for physical activity assessment [30].

Anthropometric measurements were obtained by trained personnel and according to standardized procedures [31]. Participants were weighed to the nearest 0.1 Kg using calibrated equipment (Inbody 3.0, Biospace Co. Ltd., Seoul, Korea). Standing body height (in cm) was measured to the nearest 0.5 cm with a portable wall stadiometer (Seca 213, Hamburg, Germany). A plastic, non-elastic measuring tape (Seca 201, Hamburg, Germany) was used to measure waist circumference to the nearest 0.5 cm. All anthropometric measurements were taken twice and the average of the two values was recorded. Body mass index (BMI) was calculated as weight divided by height squared (Kg/m^2^). Percent body fat was estimated using the Bioelectrical Impedance Analysis (BIA) technique (Inbody 3.0, Biospace Co. Ltd., Alpha-Tec s.a.r.l., Seoul, Korea).

Sitting systolic and diastolic blood pressure measurement were obtained twice, at 10-min intervals, using a digital sphygmomanometer. After blood withdrawal, fasting plasma glucose levels were measured using an enzymatic method (Cobas 6000, Roche, Indianapolis, IN, USA), and triglycerides, HDL-C, and low-density lipoprotein cholesterol (LDL-C) levels were determined by an enzymatic spectrophotometric technique using Vitros 350 analyzer (Ortho-Clinical Diagnostics, Johnson & Johnson, High Wycombe, United Kingdom). Plasma insulin levels were measured using radio-immunoassay (Cisbio, Codolet, France). The harmonized definition of the International Diabetes Federation (IDF) [2] was used to identify the MetS; participants were classified as having the MetS if they had three of the following five risk factors: elevated triglyceride level (≥150 mg/dL), low HDL level (<40 mg/dL for men and <50 mg/dL for women), elevated blood pressure (systolic ≥130 and/or diastolic ≥85 mm Hg), elevated FBG level (≥100 mg/dL), and elevated waist circumference (≥94 cm for men and ≥80 cm for women). In addition, high percent body fat was defined as ≥25% for men and ≥ 32% for women [31].

#### 2.2.2. Dietary Intake Assessment and Calculation of GI and GL Values

Dietary intake assessment was performed using an 86-item, semi-quantitative, and culture-specific FFQ, assessing food intake during the last 12 months before the interview [23,27,32]. Participants were asked to record the frequency of their food and beverage consumption per day, week, month, year, or never. They had the choice to report their intakes either in grams or in terms of a reference portion size. In order to assist participants in portion estimation, a standard two-dimensional food portion visual chart [33] was used, in addition to common household measures.

For data entry, a database application using Microsoft Access (Microsoft Corp., Redmond, WA, USA) was developed. For each food item or beverage, the frequency of consumption as reported by the individual was converted to daily intake. The Nutritionist Pro 1.2 software (Axxya Systems LLC, Stafford, TX, USA) was used for the estimation of energy and macronutrient content. For culture-specific/traditional food items not included in the Nutritionist Pro database, recipes were added based on a local cookbook [34]. Energy and macronutrients were calculated per gram for each food item on the FFQ list. Individual daily energy intake was then computed by summation of the respective products of the quantity consumed and the energy per gram value for each food item [35]. The same procedure was used to determine the daily intake of each macronutrient [36].

For the calculations of dietary GI and GL, all foods constitutive of the FFQ list were linked to available GI/GL values based on the literature. In particular, the International Table of GI and GL values by Foster-Powell and colleagues [37] was used. One by one, food items were manually matched to their corresponding equivalent. If the food lacked a direct match, a closely related/identical match was chosen from the table. If the food item was a mixed traditional dish, its recipe was broken down into single ingredients, which were assigned GI values from the table. Then, a GI for the whole dish was calculated using GI values of its individual foods, weighted according to each food’s carbohydrate contribution to the meal [13,38,39]. The same was applied for GL.

For the calculation of each participant’s dietary GI and GL, two approaches were used. The first approach assumes, as per the International Table of GI and GL values [37], that low-carbohydrate foods do not contribute to dietary GI/GL. Indeed, the International Table does not provide GI/GL values for low carbohydrate items such as cheese, vegetables, fish, eggs, olives, butter, alcohol, etc. This table was adopted as a reference by several studies for the estimation of the population’s dietary GI and GL [20,40,41,42,43,44,45]. Accordingly, each food item from the FFQ was assigned a GI 1 value and a GL 1 value from the table (Approach 1).

Another approach is, however, suggested by the literature [46,47,48], whereby low carbohydrate items that are excluded from the International Table (Approach 1) are taken into account in the GI/GL calculations. Several studies have adopted this rationale and included low carbohydrate food items in the analysis to better reflect the overall GI of the diet [46,47,48]. In order to apply this approach, we complemented the International Table GI values with those suggested in the literature [46,48,49]. For most of these low-carbohydrate foods, GL values were not provided and had to be calculated as follows: using Nutritionist Pro, the available carbohydrate (CHO) content of the food item, defined as the CHO that is digested, absorbed, and metabolized, was calculated [12]. This was done by subtracting total fiber from total CHO content. The CHO value was then multiplied by the corresponding GI and divided by 100 to yield the food’s GL value. Accordingly, each food item from the FFQ was assigned a GI 2 value and a GL 2 value (Approach 2).

Daily dietary GI was calculated as the sum of the GI of all foods consumed per day, multiplied by the corresponding CHO content per serving, divided by the total daily CHO consumed [50]. Daily dietary GL was calculated as the sum the GI of all foods consumed per day, multiplied by the corresponding CHO content per serving, divided by 100 [46]. These calculations are shown below:
$\mathrm{Daily}\;\mathrm{Dietary}\;\mathrm{GI}\;=\;\sum_{i=1}^{n}\mathrm{GI}i\;\times \;\mathrm{CHO}i/\sum_{i=1}^{n}\mathrm{CHO}i$$\mathrm{Daily}\;\mathrm{Dietary}\;\mathrm{GL}\;=\;\sum_{i=1}^{n}\mathrm{GI}i\;\times \;\mathrm{CHO}i/100$
where: GI*_i_* is the GI for food i, CHO_i_ is the CHO content in food i (grams per day), and *n* is the number of foods eaten per day [51]. All calculations were done for both approaches, providing each participant with daily dietary GI 1, dietary GL 1 (Approach 1), and daily dietary GI 2 and dietary GL 2 (Approach 2).

### 2.3. Statistical Analysis

Frequencies, means, and standard deviations (SD) for sociodemographic and lifestyle characteristics, anthropometric measurements, biochemical indices, cardiometabolic risk factors, and dietary intake were calculated for the study sample across categories of MetS status. Independent student t-tests were used to compare continuous variables while Chi-square and Fisher exact tests were used to compare categorical variables.

Daily dietary GI and GL were calculated and participants were grouped into quartiles of GI and quartiles of GL.

The associations between GI, GL, the MetS, and each of its abnormalities were examined using logistic regression analyses. In the logistic regression models, the MetS or its individual components were the dependent variables and GI or GL quartiles were the independent variables. Adjustment for potential confounders was performed: variables found to be significantly associated with MetS in the univariate analyses or suggested by the literature as possible confounders were adjusted for in the analyses [16]. Statistical analysis was performed using the Statistical Analysis Package for Social Sciences IBM SPSS Statistics 20 (SPSS Inc., Chicago, IL, USA). All analyses were two-tailed, and a *p* value <0.05 was considered statistically significant.

## 3. Results

The GI and GL values for food items included in the FFQ are shown in Appendix A (Approach 1 and 2). The sociodemographic and lifestyle characteristics of the study sample are presented by MetS status in Table 1. Participants’ mean age was 40.9 ± 13.7 years, with those having MetS being significantly older (*p* < 0.05). The study sample consisted of 92 (32.5%) males and 191 (67.5%) females. Within the MetS group, the proportion of females was significantly higher (54.9%) as compared to males (45.1%). More than half of the study population had an education level up to intermediate, with only 14.7% having attained university level. Those having the MetS were less likely to reach higher levels of education compared to those without the MetS. No significant differences were observed in marital status, income, or lifestyle characteristics such as smoking, alcohol, sleep difficulties, and physical activity. However, a significant difference was observed between the two groups for sedentary behavior: participants with MetS spent 307.4 ± 166.8 min per day being sedentary as compared to 263.6 ± 176.6 for participants without MetS (*p* < 0.05).

Anthropometric characteristics, biochemical, and blood pressure data are also shown in Table 1 for the study sample. BMI was significantly higher in subjects having MetS as compared to those without MetS (31 ± 5.4 vs 26.4 ± 5). As expected, this was also the case of percent body fat and waist circumference (38.7 ± 10.1 vs 34.5 ± 10.1 and 100.6 ± 11.4 vs 87.1 ± 12, respectively). Biochemical values (including total cholesterol, LDL cholesterol, insulin, and glycated hemoglobin (HbA1c)) and both systolic and diastolic blood pressure were all significantly higher in the MetS group compared to those without the MetS (*p* < 0.05).

Dietary energy and macronutrient intakes as well as daily dietary GI and GL estimates are displayed in Table 2 for the study population. Participants with MetS had higher intakes of energy (Kcal/day), carbohydrate (grams per day and percent calories), total sugars (grams per day), and fat (grams per day and percent calories) but these differences did not reach statistical significance. In addition, participants with MetS had higher dietary GI and GL estimates (in both approaches), with the differences being borderline significant (*p* = 0.053, 0.050, and 0.058 for GI 1, GL 1, and GL 2, respectively). In addition, dietary energy and macronutrient intakes of participants were examined across GI quartiles, as shown in Appendix A.

Among participants with and without the MetS, a notable difference in distribution was observed across quartiles of GI and GL as shown in Figure 1 and Appendix A.

As indicated earlier, the final sample for the present study included 283 subjects. Given that among subjects with no MetS, 20% belonged to Quartile 4 of GI, the availability of 283 subjects would allow for the detection of an odds ratio (OR) of 1.2 with a 5% error and 96% power [16].

In the present study, the association between GI 1, GL 1, and MetS and its components were examined using several logistic regression models (Table 3 and Appendix A), including the crude unadjusted model; model 1 (adjusted for age and gender); model 2 (adjusted for age, gender, BMI, smoking status, alcohol intake, energy intake, total fiber intake, sedentary behavior, and education level); and model 3 for GI only (adjusted for all variables in model 2, in addition to percentage of energy from both protein and fat) [16]. In the crude model, participants belonging to the highest quartile of GI had significantly higher odds of developing MetS (OR: 2.251, 95% CI: 1.120–4.525). In the same model, participants in the highest quartile of GI had significantly higher odds of having elevated triglyceride levels (OR: 2.157, 95% CI: 1.022–4.552). However, these associations lost significance with further adjustments. It was also observed that participants belonging to the second quartile of GI had significantly lower odds of having elevated FBG (OR: 0.464, 95% CI: 0.225–0.957) in the crude model, and this association remained significant with additional adjustments in model 1 (OR: 0.377, 95% CI: 0.175–0.810), model 2 (OR: 0.380, 95% CI: 0.174–0.833), and model 3 (OR: 0.380, 95% CI: 0.174–0.833). There was no significant association between GI and the risk of elevated waist circumference or percent body fat in all the models (Appendix A). As for GL, no significant association was observed with MetS in all models (Table 3). For triglycerides, a significant association was found with the second quartile of GL in Model 2, with an OR of 0.425 (95% CI of 0.181–0.995). No other associations were detected between GL and any of the MetS risk factors or adiposity indicators as shown in Appendix A. The same regression analyses were conducted for daily dietary GI 2, GL 2, and MetS and its components, and the results did not show any significant association (data not shown).

Sensitivity analyses were performed while also including subjects with implausible energy reporters (below 500 calories or above 6000 calories), and the results were similar. In addition, all analyses were conducted after stratification by gender, and the results were similar to those obtained for the overall sample.

## 4. Discussion

To our knowledge, this is the first study from the Arab EMR to determine dietary GI and GL and investigate their association with metabolic abnormalities. We have used two different approaches for the estimation of dietary GI and GL in Lebanese healthy adults, and the results of both approaches did not show any significant association between GI, GL, and the MetS. When examining the association between dietary GI, GL, and individual components of MetS, lower odds of elevated blood glucose and elevated triglycerides were found in those belonging to the second quartile of GI and GL, respectively.

For the assessment of daily dietary GI and GL, two approaches were used in our study. The first was based on matching the foods listed in the FFQ to GI values included in the International Table by Foster-Powell and colleagues (Approach 1) [37], while in the second approach, additional literature was consulted to assign GI values for low carbohydrate foods that are not featured in the International Table (Approach 2) [46,47,48,49]. Accordingly, average dietary GI was found to range between 59.9 (Approach 1) and 61.2 (Approach 2). These estimates are in line with those reported from Iran, another country of the EMR where average GI was estimated at 63.7 [52]. Our estimates are also similar to those reported from Malaysia (55.8–58.6) [53], Australia (57.5) [44], and Brazil (55.4–55.5) [41], while being relatively higher than those reported from the USA (53.1–54.6) [20] and Mexico (51.8) [45]. Because GI is not a component of the standard output provided by nutrient analysis softwares, the developed GI database for Lebanese foods may be of use for other studies investigating diet–disease associations in the region.

In our study, the MetS prevalence was estimated at 35.6%, which is in line with the previously reported estimates of 34.6% in 2013 and 31.2% in 2007 among Lebanese adults [21,23]. Our results did not show any significant association between dietary GI and the MetS, after adjustment for potential confounders. Studies investigating such associations are lacking in the EMR, but previous studies conducted in other parts of the world yielded equivocal results [17,19,20,41,43,52,53,54]. In accordance with our findings, dietary GI was not found to be associated with the MetS among adults from São Paolo, Korea and the USA [20,41,54]. In contrast, other studies reported a positive association between GI and MetS [17] while also suggesting that this association may be age-dependent [18] or gender-specific [20]. Despite reporting a positive association between dietary GI and MetS (OR = 1.23, 95% CI: 1.10–1.38), a recent systematic review of studies conducted among healthy adults noted that there was significant evidence for publication bias: in other words, studies showing inverse or null associations may have been missing [16]. An important consideration that may modulate the association between dietary GI and MetS is the intake level of the various macronutrients as these dietary components may affect insulin resistance and other metabolic abnormalities characteristic of the MetS [14]. For instance, in an 11-week intervention among healthy overweight subjects, there was no effect of diets that differed in GI but had comparable macronutrient content, on MetS abnormalities [14]. In our study, significant differences in macronutrient intakes were in fact observed across quartiles of GI (Appendix A). More specifically, those belonging to Q1 of dietary GI had significantly lower carbohydrates intake, while having the highest intakes in terms of total fat (41.9% energy intake) and saturated fat (11.4% energy intake) (Appendix A). High fat intake and particularly high saturated fat consumption may affect insulin sensitivity, lipid profile, and glycemic control [55] independently from carbohydrates. Participants belonging to Q1 of dietary GI also had the highest intakes of total sugars (16.3%). This finding may seem counter-intuitive but several studies have previously shown that the main contributors to dietary GI are not sugars but rather staple starches and refined grains [56,57].

Contrary to the expectations, we found a decreased risk of elevated FBG in those belonging to the second quartile of dietary GI, compared to those in the first, and this association persisted after adjustment for potential confounder. Previous studies have reported inconclusive findings related to the association between GI and fasting glycemia. Few studies have suggested an increase in FBG with increasing dietary GI [58], while others have argued that beneficial effects of lower GI on FBG can be shown only in subjects who already have impaired glucose tolerance or who already have overt Type II Diabetes (T2D) [59]. Additionally, our findings showed that high GI was associated with significantly increased risk for elevated triglycerides, but this association lost significance after adjusting for confounders. The association between GI and elevated triglycerides has been reported by several previous studies [20,60], while, others, as is the case with our study, did not observe such an association [41,61,62]. In a trial conducted among non-diabetic adults, Vrolix et al. [14] did not observe any significant effect of dietary GI on fasting serum TAG concentrations, which agrees with most other intervention studies in nondiabetic subjects [63].

In accordance with several previous studies [19,20,51], daily dietary GL for Lebanese adults was calculated as the product of the GI of the consumed foods and the corresponding carbohydrate content per serving. Accordingly, average dietary GL estimates were found to range between 209.7 (Approach 1) and 213.9 (Approach 2). Our estimates are relatively higher than those reported by studies conducted in Brazil (108.5–111.5) [41], Iran (104.8–163.6) [52], and the USA (114.6–145.2) [20] as well as those reported by the PREDIMED study (107.4–113.2) [18]. This could be due to the fact that GL is a quantitative indicator and dietary assessment in our study was conducted using an FFQ, which tends to overestimate dietary intake [64,65,66,67]. The previously mentioned studies used different dietary assessment tools including the 3-day diet record [20] and 24-h recall [19], which may affect dietary GL assessment. Our dietary GL estimates are similar to those reported by Kaur et al. (2018) in Malaysia where average GL was reported to range between 185 and 213 and where, similarly to our study, dietary assessment was performed using a FFQ [53]. Our study findings did not reveal an association between GL, the MetS, or any of its components, except for a lower risk of hypertriglyceridemia in Q2 compared to Q1. Several other studies have also reported null associations between dietary GL and the MetS in adults [18,19], or have unexpectedly reported an inverse association between GL and the MetS [20]. A recent systematic review and dose–response meta-analysis concluded that available evidence does not support a significant association between GL and the prevalence of the MetS in adults [16].

Despite the growing interest in GI and GL as markers of risk factors for disease, the methods for assessing these exposures in an epidemiologic context are neither well established nor consistently applied [42]. It is recognized that the GI value of a food may be subject to considerable variations depending on the extent of ripeness, processing, cooking method, extent of starch gelatinization, and storage duration [16,68]. Other concerns related to dietary GI/GL estimation are whether foods consumed together may have an impact on each other to alter the GI/GL of the whole meal [16,68,69]. While some authors suggest that the GI of a meal can be calculated by adding the carbohydrate contributions of each constituent food multiplied by its published GI [13,70], another school of thought argues that a food is more than just the sum of its nutrients due to several chemical and physical interactions that may occur. Combining macronutrients was found to influence GI, the latter being positively associated with carbohydrate content and negatively associated with protein and fat content, which can significantly reduce the glycemic response [71]. Foster-Powell and colleagues suggest that foods should be tested in the geographical area where they are consumed [37].

The present study had several strengths. It was conducted on a random sample of the adult population living in Beirut and was performed using a well-planned design, protocol, and methodology [26,27,28]. In our study, subjects who had previous diagnosis of chronic diseases or metabolic abnormalities were excluded to decrease potential reverse causation. In addition, we collected information on several lifestyle factors, including smoking, alcohol drinking, and sedentary behavior, which allowed us to adjust for potential confounders when examining the association between GI/GL and MetS. However, the results of this study ought to be interpreted in light of the following limitations. First, this is an exploratory study that adopted a cross-sectional design, which means that the generated findings only reflect the status at the time of the investigation. In addition, despite having used the International GI/GL Table, which is the most commonly used source of GI values [37], it is important to acknowledge that this table has its own set of limitations such as broad groupings, multiple entries, missing values, and laboratory errors [37,42]. Similarly, the International Table of GI and GL values does not provide GI values for all foods, especially for cultural and traditional foods. We have assigned GI/GL values of similar foods to our traditional foods or estimated GI/GL based on recipes, which may introduce an error in dietary GI and GL calculations [16]. Acknowledging that dietary fiber may affect the GI of foods, we have adjusted for total dietary fiber intake in the regression analyses, but the type of fiber was not taken into account. In addition, in our analyses, we have adjusted for the percent contribution of protein and fat, but this may not account for the physical interactions that may occur between the various components of the meal and hence alter its GI/GL. Like other studies where dietary intake is assessed by means of self-exposure, dietary assessment performed in our study may be prone to measurement errors [16]. Measurement error in the estimation of dietary GI and GL may result in the attenuation of true associations in an observational study [16]. The use of the FFQ approach may also carry some inherent limitations such as the individuals’ ability to estimate the frequency and portion sizes of their usual dietary intake [72], and the fact that the food list itself may not include all the carbohydrate-containing foods that a specific person consumes. However, despite these potential limitations, the FFQ was described as “the most robust method for assessing an individual’s average dietary intake compared with other assessment methods in large studies” [16]. Although the FFQ that was used in this study was not previously validated, it has been used in several studies, yielding plausible results [23,27,32]. Given that the questionnaire was filled in an interview setting, this approach may be associated with social desirability bias, whereby participants may respond in a way that they believe is acceptable or favorable to the interviewer [27,73]. In our study, field workers who performed data collection underwent extensive training to decrease any judgmental verbal or non-verbal communication and thus to minimize social desirability bias. It is also important to note that this study was restricted to the urban setting of the Greater Beirut area and, hence, findings related to food consumption dietary GI/GL may not be representative of less urban areas in the country and future nationally representative studies are needed. Finally, the potential of multiple testing in our study should be acknowledged and considered when interpreting the findings.

Despite its limitations, the present study contributes to the body of evidence discussing the relationship between GI, GL, and MetS, in a nutrition transition context. It showed that, among healthy Lebanese adults, GI and GL were not associated with increased risk of MetS in a cross-sectional design. Future studies, including high-quality cohort studies and clinical trials, should be conducted in the region with the aim of achieving a higher level of evidence on the association between GI/GL and cardiometabolic risk while also elucidating the potential causal pathways between them.

## Figures and Tables

**Figure 1 nutrients-12-01394-f001:**
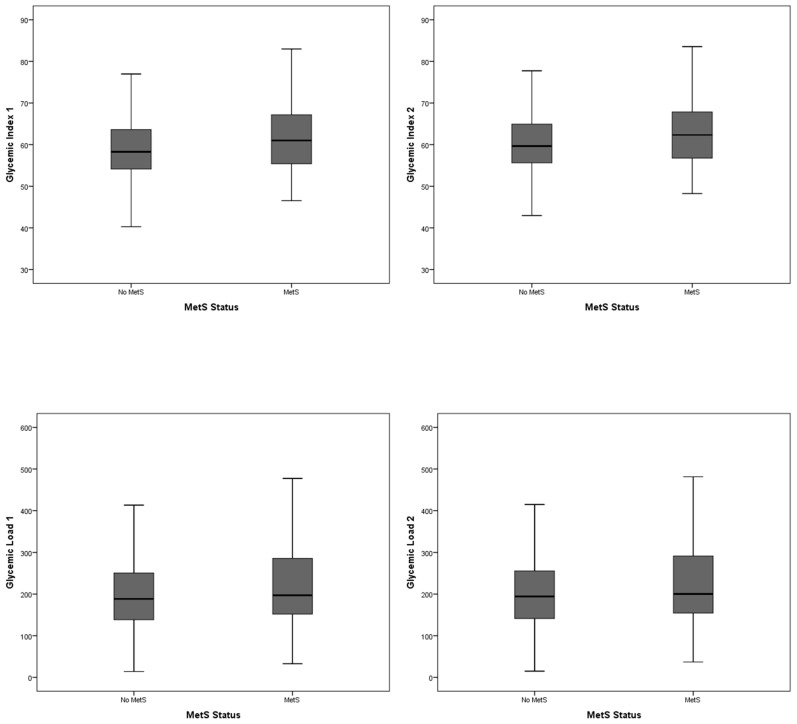
Box plot showing the distribution of dietary GI and GL among participants with and without MetS. GI: Glycemic index; GL: Glycemic load; MetS: Metabolic syndrome.

**Table 1 nutrients-12-01394-t001:** Sociodemographic, lifestyle, and anthropometric characteristics and cardiometabolic risk factors in the study population (*n* = 283) of Lebanese adults and by MetS status.

	All Participants (*n* = 283)		Participants without MetS (*n* = 181)		Participants with MetS (*n* = 102)		
Variables	Mean or *n*	SD or %	Mean or *n*	SD or %	Mean or *n*	SD or %	*P* value
**Age (years)**	40.9	13.7	38.8	12.7	44.8	14.6	*p* < 0.001
**Gender**							*p* = 0.001
Male	92	32.5	46	25.4	46	45.1	
Female	191	67.5	135	74.6	56	54.9	
**Marital Status**							*p* = 0.344
Single	91	31.8	54	29.8	36	35.3	
Married	195	68.2	127	70.2	66	64.7	
**Income/month**							*p* = 0.164
<600$	75	28.1	42	25.6	33	33	
600$–2000$	167	62.5	105	64	60	60	
>2000$	25	9.4	17	10.4	7	7	
**Education**							*p* = 0.006
Elementary to intermediate	147	55.5	87	51.2	59	64.1	
Secondary or technical	79	29.8	49	28.8	28	30.4	
University	39	14.7	34	20	5	5.4	
**Physical Activity**							
Total minutes per day	110.3	81.5	113.9	85.8	103	73.3	*p* = 0.327
Sedentary (minutes/day)	279.3	174.8	263.6	176.6	307.4	166.8	*p* = 0.043
**Smoking**							*p* = 0.657
No	63	22.3	42	23.2	21	20.6	
Yes	220	77.7	139	76.8	81	79.4	
**Alcohol Consumption**							*p* = 0.187
No	196	69.3	115	85.6	81	79.4	
Yes	47	16.6	26	14.4	21	20.6	
**Sleeping Difficulties**							*p* = 0.259
No	168	59.4	112	61.9	56	54.9	
Yes	115	40.6	69	38.1	46	45.1	
**Anthropometric Characteristics**							
BMI (Kg/m^2^)	28	5.61	26.4	5	31	5.4	*p* < 0.001
Percent Body Fat (%)	36	10.3	34.5	10.1	38.7	10.1	*p* = 0.001
Waist circumference (cm)	91.9	13.5	87.1	12	100.6	11.4	*p* < 0.001
**Biochemical and Blood Pressure Data**							
Total Cholesterol (mg/dL)	172.5	13.4	178.1	36.5	192.7	43.2	*p* = 0.003
LDL-C (mg/dL)	99.5	14.8	101.8	31.4	116.2	38.3	*p* = 0.001
Triglycerides (mg/dL)	115	60.8	96.1	50.9	164.4	80	*p* < 0.001
HDL-C (MG/DL)	50.5	10.6	57	16	42.9	10.9	*p* < 0.001
**Blood Pressure**							
SBP (mmHg)	101.2	12.4	111.7	13.2	125.5	18.4	*p* < 0.001
DBP (mmHg)	65	7	70.3	8.2	77.4	10.3	*p* < 0.001
**Measures of Glycemia**							
Fasting blood glucose (mg/dL)	98.2	13.3	94	7.2	105.7	17.7	*p* < 0.001
HbA1c (%)	5.5	0.5	5.3	0.4	5.7	0.6	*p* < 0.001
Insulin (μU/mL)	26.3	15.5	23.4	8.8	31.4	22.1	*p* < 0.001

MetS: Metabolic syndrome; BMI: Body Mass Index; LDL-C: Low-Density Lipoprotein-Cholesterol; HDL-C: High-Density Lipoprotein-Cholesterol; SBP: Systolic Blood Pressure; DBP: Diastolic Blood Pressure; HbA1c: glycated hemoglobin.

**Table 2 nutrients-12-01394-t002:** Dietary energy, macronutrients, GI, and GL of participants with and without MetS.

	All Participants (*n* = 283)	Participants without MetS (*n* = 181)	Participants with MetS (*n* = 102)	Significance
Mean ± SD				
**Energy (Kcal/day)**	3131.2 ± 1302.6	3080.2 ± 1281.6	3232.1 ± 1337.9	0.347
**Protein (g/day)**	102.7 ± 3.6	103.3 ± 65.8	101.9 ± 50.4	0.854
**Protein (% of energy)**	13 ± 3.6	13.2 ±3.9	12.7 ± 3.2	0.224
**Fat (g/day)**	131.8 ± 64.8	130.1 ± 63.8	134.8± 67.3	0.560
**Fat (% of energy)**	39.1 ± 7.9	39.4 ± 7.7	38.6 ± 8.1	0.385
**Carbohydrates (g/day)**	387.55 ± 158.4	377.4 ± 150.4	407.4 ± 170.2	0.126
**Carbohydrate (% of energy)**	50.3 ± 8.3	50 ± 8.2	51 ± 8.4	0.360
**Total Sugar (g/day)**	105 ± 58.5	101.5 ± 56.3	111.2 ± 61.8	0.181
**Total Sugar (% of energy)**	13.8 ± 6	13.9 ± 6.4	13.6 ± 5.3	0.689
**Dietary Fibers (g/day)**	28.1 ± 11.8	28.7 ± 13.5	27.8 ± 10.7	0.563
**Glycemic Index 1 ^a^**	59.9 ± 8	59.2 ± 7.8	61.2 ± 8.2	0.053
**Glycemic Index 2 ^b^**	61.2 ± 7.8	60.6 ± 7.6	62.3 ± 7.9	0.076
**G1ycemic Load 1 ^a^**	209.7 ± 100.3	201.5 ± 95.8	225.8 ± 106.2	0.050
**Glycemic Load 2 ^b^**	213.9 ± 101.2	205.9 ± 97	229.6 ± 106.8	0.058

GI: Glycemic index; GL: Glycemic load; MetS: Metabolic syndrome. ^a^ Values based on Approach 1 (International table): considering only carbohydrate-rich foods [37]. ^b^ Values based on Approach 2: same as Approach 1 in addition to GI and GL values proposed by studies (Schulz et al., 2005; van Bakel et al., 2009) [46,48] and USDA CSFII 94-96 food codes [49] with the help of NutritionistPro records at the American University of Beirut (AUB).

**Table 3 nutrients-12-01394-t003:** Multivariable logistic regression analyses of MetS by dietary GI 1 and GL 1 quartiles.

	Quartile 1 (*n* = 71)	Quartile 2 (*n* = 72)	Quartile 3 (*n* = 72)	Quartile 4 (*n* = 71)
	OR (95% CI)			
	Daily Glycemic Index 1			
Crude model	1	1.225 (0.600–2.503)	1.251 (0.612–2.559)	2.251 (1.120–4.525)
Model 1 ^a^	1	1.093 (0.517–2.311)	1.138 (0.539–2.402)	1.483 (0.702–3.134)
Model 2 ^b^	1	1.258 (0.547–2.891)	1.090 (0.473–2.512)	1.269 (0.546–2.945)
Model 3 ^c^	1	1.195 (0.518–2.756)	0.973 (0.414–2.289)	1.215 (0.518–2.847)
	Daily Glycemic Load 1			
Crude model	1	1.432 (0.711–2.885)	1.027 (0.502–2.101)	1.965 (0.981–3.936)
Model 1 ^a^	1	1.330 (0.638–2.774)	0.672 (0.304–1.485)	1.572 (0.710–3.480)
Model 2 ^b^	1	0.941 (0.407–2.173)	0.579 (0.236–1.421)	1.595 (0.657–3.875)

MetS: Metabolic syndrome; GI: Glycemic index; GL: Glycemic load; OR: Odds ratio; CI: Confidence interval. ^a^ Model 1: adjusted for age and gender. ^b^ Model 2: adjusted for age, gender, BMI, smoking status, alcohol intake, energy intake, total fiber intake, sedentary behavior, and education level. ^c^ Model 3: adjusted for age, gender, BMI, smoking status, alcohol intake, energy intake, total fiber intake, sedentary behavior and education level and percentage of energy from protein and fat. Significant results are shown in bold.

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
