# Peer review of "Dietary Glycemic Index and Glycemic Load Are Not Associated with the Metabolic Syndrome in Lebanese Healthy Adults: A Cross-Sectional Study"

_nutrients, 2020, doi:10.3390/nu12051394_

Round 1

Reviewer 1 Report

Thank you for allowing me to Review this interesting paper on GI and GL in relation to MetS.

This is a well written paper on well performed research on the important topic of the identification and prevention of metabolic syndrome.

Of special interest is the socio-economic context and the general medical status of the paper, as is well outlined in Table 1. Although self reported (an issue well addressed on page 17, line 414 and 415), the Appendices 1 and 2 are also very informative as they provide insight in daily life of participants.

Several suggestions:

ABSTRACT.

The number of participants, total and according to group, must be mentioned. The authors describe well how they get to their numbers (line 100 - 111, page 3, paragraph 2.1.)

BODY OF MANUSCRIPT.

The paper is (very) well written and the project has enough data and is of sufficient interest for a complete thesis, however for a Nutrients publication I would suggest the text may be shortened where ever possible.  Please refer with results to the Tables if applicable.

Somewhere there needs to be a small note that the authors are aware of the exploratory nature of the project and the issue of multiple testing has passed the mind.  

TABLES.

Very interesting Table 1. In the text the blood pressure issue is well addressed.

Table 2 nicely resumes nutritional information.

Table 3 overlaps with and details the GI and GL results of Table 2. The authors may creatively consider to convert Table 3 to a graph, for example a box-and-whiskers plot of the GI and GL quartiles, or alternatively a box plot (with the absolute numbers included in the boxes).

Turning the Table 3 (which just shows numbers) into a graph may address and illustrate the biological relevant consequences of the GI and GL findings better then the statistically calculated (with a borderline statistical significant mainly due to subgrouping) impacts of GI and GL. A graph of the Table 3 data would also addess the distribution issue mentioned on page 8, line 249 - 251. With the graph in the main paper, Table 3 could be moved to the Supp. material, or alternatively (the last 4 lines of) Table 2 could be extended.

Consider Table 4 and 5 to be moved to a Supplementary section. The tables are well described in the text (this text may remain), but may be of interest to a small part of selected statistically interested readers only.

Hope this helps.

Author Response

We would like to thank the reviewer for all the constructive comments provided on the manuscript. Please find below a point-by-point reply showing how we have dealt with  each of the comments:

Comment 1: ABSTRACT.

The number of participants, total and according to group, must be mentioned. The authors describe well how they get to their numbers (line 100 - 111, page 3, paragraph 2.1.)

Response to Comment 1:

Thank you for your constructive comments and feedback.

Line 23 in the Abstract has been amended to include the total number of the original survey: “The study uses data from a community-based survey of 501 Lebanese urban adults”. (Page 1)

Moreover, lines 26-28 have been revised as follows: “Subjects with previous diagnosis of chronic disease, metabolic abnormalities, or with incomplete data or implausible energy intakes were excluded, yielding a sample of 283”.

 Comment 2: BODY OF MANUSCRIPT.

The paper is (very) well written and the project has enough data and is of sufficient interest for a complete thesis, however for a Nutrients publication I would suggest the text may be shortened where ever possible. 

Response to Comment 2:

Based on the reviewers’ suggestion we have revised the manuscript and shortened several sections.

 Comment 3: Please refer with results to the Tables if applicable.

Response to Comment 3:

As advised by the reviewer, we made sure to refer to the tables when applicable.

Comment 4: Somewhere there needs to be a small note that the authors are aware of the exploratory nature of the project and the issue of multiple testing has passed the mind. 

Response to Comment 4:

Based on the reviewer’s recommendation, we have amended the limitations section to include the exploratory nature of the study and to acknowledge the possibility of multiple testing, as follows:

Lines 409-410: “First, this is an exploratory study that adopted a cross-sectional design, which means that the generated findings only reflect the status at the time of the investigation.” (Page 16)

Lines 442-443: “Finally, the potential of multiple testing in our study should be acknowledged and considered when interpreting the findings.” (Page 16)

Comment 5: TABLES.

Very interesting Table 1. In the text the blood pressure issue is well addressed.

Table 2 nicely resumes nutritional information.

Table 3 overlaps with and details the GI and GL results of Table 2. The authors may creatively consider to convert Table 3 to a graph, for example a box-and-whiskers plot of the GI and GL quartiles, or alternatively a box plot (with the absolute numbers included in the boxes).Turning the Table 3 (which just shows numbers) into a graph may address and illustrate the biological relevant consequences of the GI and GL findings better then the statistically calculated (with a borderline statistical significant mainly due to subgrouping) impacts of GI and GL. A graph of the Table 3 data would also address the distribution issue mentioned on page 8, line 249 - 251. With the graph in the main paper, Table 3 could be moved to the Supp. material, or alternatively (the last 4 lines of) Table 2 could be extended.

Response to Comment 5:

As suggested by the reviewer, Table 3 has been converted to a box-plot graph (Figure 1) to better illustrate the findings. Moreover, Table 3 has been moved to the Supplementary File, as Table S3.

The text was also amended as follows:

“Among participants with and without the MetS, a notable difference in distribution was observed across quartiles of GI and GL as shown in Figure 1 and Supplementary Table S3.” (Lines 261-264, Page 9)

Comment 6: Consider Table 4 and 5 to be moved to a Supplementary section. The tables are well described in the text (this text may remain), but may be of interest to a small part of selected statistically interested readers only.

Response to Comment 6:

Based on the reviewer’s recommendation, we have moved the tables showing association between GI, GL and individual metabolic abnormalities to the Supplementary File (Tables S4 and S5), but we kept the association between GI, GL and the MetS in one short table in the manuscript given that it addresses one of the main objectives of the study. (Table 3) (Page 13)

The text has been amended to properly refer to the tables. (Line 280, Page 12)

Reviewer 2 Report

My comments are highlighted in the attached file.

Author Response

Reviewer #2:

We would like to thank the reviewer for all the constructive comments and recommendations to improve our manuscript. Please find below a point-by point response to each of the comments:

Comment 1: Line 69: PREDIMED.

Response to Comment 1:

Thank you. The typo in Line 71 has been corrected. (Page 2)

Comment 2: Lines 163-164: It is not completely explained… How did you value items that were consumed weekly, monthly or even a once a year for total energy and macronutrients quantification? 

Response to Comment 2:

Based on the reviewer’s comment, we have added the following clarification:

Lines 163-165: “For data entry, a database application using Microsoft Access (Microsoft Corp., Redmond, WA, USA) was developed. For each food item or beverage, the frequency of consumption as reported by the individual was converted to daily intake.” (Page 4)

This means that if, for example, the frequency was expressed on a weekly basis, the database application will divide it by 7 to determine its daily intake.

We have also explained the way the quantification of macronutrients was performed, as follows: Lines 169-172: “Energy and macronutrients were calculated per gram for each food item on the FFQ list. Individual daily energy intake was then computed by summation of the respective products of the quantity consumed and the energy per gram value for each food item. The same procedure was used to determine the daily intake of each macronutrient.” (Page 4)

 Comment 3: Line 197: Add to the formula "divided by 100".

Response to Comment 3:

Thank you for alerting us to this. The formula has been edited as such: Daily Dietary GL = / 100 (Line 207, Page 5)

Comment 4: Lines 244-245: I think that you should mentioned here that you evaluate the % of saturated fatty acid, % of monounsaturated fatty acid, etc and did not find any differences (link this information to the supplemental material).

Response to Comment 4:

Based on the reviewer’s comment we have added the following to the Results section (Lines 259-260): “In addition, dietary energy and macronutrient intakes of participants were examined across GI quartiles, as shown in Supplementary Table S2.” (Page 9)

Comment 5: Line 327: São Paulo.

Response to Comment 5:

Thank you. The typo has been corrected. (Line 333, Page 14)

Comment 6: Line 332: Did you considered to stratify your results by gender, as you have twice women than men? You explain this in your discussion, but have you tried to perform that analysis?

Response to Comment 6:

As rightfully pointed by the reviewer and given that we had twice women as much as men, we have performed all analyses by gender. The results obtained were similar to those obtained for the overall sample. All the results and tables can be shared with the reviewer if needed. 

To reflect the above, we have amended the results section, as follows: “In addition, all analyses were conducted after stratification by gender, and the results were similar to those obtained for the overall sample.” (Lines 298-300, Page 12)        

Comment 7: Lines 343-344: Did you evaluate sugar intake as well? It was not mentioned in the supplemental material, but I think that is as important as fiber intake in this context.

Response to Comment 7:

Thank you for pointing this out. We have added the data on sugar to Tables 2 and Supplementary Table S2.

Moreover, the text has been amended as such: “Participants with MetS had higher intakes of energy (Kcal/day), carbohydrate (grams per day and percent calories), total sugars (grams per day), and fat (grams per day and percent calories) but these differences did not reach statistical significance.” (Lines 254-256, Page 9)

We have also added the following to the discussion: “Participants belonging to Q1 of dietary GI had also the highest intakes of total sugars (16.3%). This finding may seem counter-intuitive but, several studies have previously shown that the main contributors to dietary GI are not sugars but rather staple starches and refined grains [54,55].” (Lines 353-355, Page 14)

The references [54,55] mentioned above are the following:

Sluik, D.; Atkinson, F.S.; Brand-Miller, J.C.; Fogelholm, M.; Raben, A.; Feskens, E.J. Contributors to dietary glycaemic index and glycaemic load in the Netherlands: the role of beer. Br J Nutr 2016, 115, 1218-1225.

Kusnadi, D.T.L.; Barclay, A.W.; Brand-Miller, J.C.; Louie, J.C.Y. Changes in dietary glycemic index and glycemic load in Australian adults from 1995 to 2012. Am J Clin Nutr 2017, 106, 189-198.

Comment 8: Appendix 2: And sugars?

Response to Comment 8:

We have addressed this comment, as explained above.